# The Feasibility of Liver Biopsy for Undefined Nodules in Patients under Surveillance for Hepatocellular Carcinoma: Is Biopsy Really a Useful Tool?

**DOI:** 10.3390/jcm11154399

**Published:** 2022-07-28

**Authors:** Matteo Renzulli, Anna Pecorelli, Nicolò Brandi, Stefano Brocchi, Francesco Tovoli, Alessandro Granito, Gianpaolo Carrafiello, Anna Maria Ierardi, Rita Golfieri

**Affiliations:** 1Department of Radiology, IRCCS Azienda Ospedaliero-Universitaria di Bologna, Via Albertoni 15, 40138 Bologna, Italy; pecorelli.anna@gmail.com (A.P.); stefano.brocchi85@gmail.com (S.B.); rita.golfieri@unibo.it (R.G.); 2Division of Internal Medicine, IRCCS Azienda Ospedaliero-Universitaria di Bologna, 40138 Bologna, Italy; francesco.tovoli2@unibo.it (F.T.); alessandro.granito@unibo.it (A.G.); 3Diagnostic and Interventional Radiology Department, Fondazione IRCCS Cà Granda Ospedale Maggiore Policlinico, Università degli Studi di Milano, 20021 Milan, Italy; gcarraf@gmail.com (G.C.); amierardi@yahoo.it (A.M.I.)

**Keywords:** hepatocellular carcinoma, liver biopsy, feasibility, liver imaging, multidisciplinary meeting

## Abstract

**Background**: The aim of the present study is to determine the feasibility of biopsy for atypical liver nodules in patients under surveillance for hepatocellular carcinoma (HCC), assessing which factors influence the decision to perform it. **Methods**: A total of 128 atypical liver nodules in 108 patients under surveillance for HCC, who underwent CT between September 2018 and September 2019, were included. All the images were saved digitally (on CD-ROM) and the two most representative images in the arterial and delayed phases were selected for each lesion and inserted into a digital atlas (on PDF). Two experienced radiologists (Readers 1 and 2) reviewed both the CD-ROM and the PDF to define the feasibility of biopsy in both scenarios, specifying the reasons for the unfeasibility of biopsy. The intra-observer variability and inter-observer variability were assessed. **Results**: When reviewing the PDF, 76 (59.4%) and 68 (53.1%) nodules were deemed unfeasible for biopsy by the less experienced radiologist (Reader 1) and the more experienced radiologist (Reader 2), respectively (*p* = 0.604). When reviewing the entire CT study, both percentages decreased slightly (Reader 1 = 70/128 (54.7%); Reader 2 = 61/128 (47.6%); *p* = 0.591). The intra-reader agreement on the PDF was substantial (k = 0.648 (95% CI = 0.513–0.783)). The inter-reader agreement on the PDF was slight (k = 0.185 (95% CI = 0.021–0.348)) and moderate on the entire CT study (k = 0.424 (95% CI = 0.269–0.579)). When assessing the PDF, the nodule size (10–20 mm) and location in segments six and eight were negatively and positively associated with the feasibility of liver biopsy, respectively. When assessing the CD-ROM, only the nodule dimension was associated with the unfeasibility of liver biopsy. **Conclusions**: The unfeasibility of liver biopsy is mainly due to the small size of the lesions and their location.

## 1. Introduction

Liver cancer is the sixth most commonly diagnosed cancer and the third commonest cause of cancer-related deaths worldwide [1]. Hepatocellular carcinoma (HCC) accounts for up to 90% of liver cancers and, in the majority of cases, it occurs in the context of fibrosis or cirrhosis, the main aetiology of which is, for the most part, viral. In recent years, the advent of vaccination against hepatitis B virus (HBV) and the diffusion of new antiviral drugs for hepatitis C virus (HCV), along with the widespread epidemic of non-alcoholic fatty liver disease (NAFLD), have slightly changed the prevalence of HCC risk factors [1]. Furthermore, new evidence suggests that, in the contexts of metabolic syndrome and NAFLD, a significant proportion of patients may develop HCC, including those with non-cirrhotic livers [2].

The guidelines of the international hepatological societies suggest surveilling patients at high risk of developing HCC in order to detect the tumour at an early stage [3,4]. Different authors have demonstrated an association between HCC surveillance and early tumour detection, with an improved survival rate in cirrhotic patients [5,6]. This is due to the fact that patients with HCC detected at an early stage are eligible for curative treatments, such as liver transplantation or surgical resection, with 5-year survival rates of approximately 70% [7,8]. Conversely, patients with more advanced tumours are eligible only for palliative therapies, with a poor prognosis (a median survival of 1 to 2 years) [9,10,11,12]. Unfortunately, ultrasound (US) detects early-stage HCC with a sensitivity of only 47% [13]. In recent years, much effort has been made to overcome the US limitations. In particular, non-enhanced magnetic resonance imaging (MRI) has been proposed as a potential surveillance tool for HCC, thanks to its higher sensitivity and specificity as compared to US [14,15,16,17]. The direct consequence of this strategy will be the need for a correct characterisation of an ever-increasing number of liver lesions.

Hepatocarcinogenesis is a multistep process, during which epigenetic and genetic alterations accumulate, leading to the progression from precancerous to overt malignant lesions [18]. These changes occur at different time points, and the nodules possibly encountered in fatty, fibrotic, or cirrhotic livers comprise large regenerative nodules (LRNs), low- or high-grade dysplastic nodules (DNs), early HCC, and overt HCC. During hepatocarcinogenesis, intra-nodular vascular changes occur with a progressive decrease in the portal blood supply and a parallel increase in the arterial flow provided by the so-called “unpaired” arteries. The vascular derangement produces the typical radiologic hallmark of HCC, that is, arterial phase hyperenhancement (APHE) and the wash-out of the contrast media in the portal and/or delayed phases. Based on this peculiar behaviour, both the European and the American guidelines suggest adopting the above as the non-invasive diagnostic criteria for patients at high risk of developing HCC, using multiphasic contrast-enhanced computed tomography (CECT) and/or dynamic contrast-enhanced MRI [3,4]. In nodules larger than 10 mm, the presence of the typical radiologic hallmark—the “wash-in” and “wash-out”—is accurate enough to reach a diagnosis of HCC. However, the sensitivity of the imaging technique for detecting HCC dramatically drops for liver lesions ranging between 10 and 20 mm [19], and it is even worse for lesions of less than 10 mm [20]. The correct diagnosis of a lesion in cirrhotic or fatty livers is of extreme importance. In fact, as previously mentioned, the patient outcome can significantly change, as the patient can be treated with the most effective procedure or avoid unnecessary and morbid treatment [21]. For these reasons, international guidelines recommend establishing pathologic proof for all cirrhotic patients who have a focal liver lesion with an atypical radiologic vascular pattern at both CT and/or MRI, as well as for all nodules arising in a non-cirrhotic liver. From this perspective, it is worth remembering that, in recent years, the wide diffusion of MRI performed with hepato-specific contrast media [22,23] has led to the detection of a notable number of hypointense nodules in the hepatobiliary phase, without arterialisation. According to the current guidelines, to date, these nodules remain undefined [24,25] and, therefore, have the possibility of being characterised only through biopsy.

Liver biopsy is a problem-solving technique in the case of atypical imaging features for HCC after the initial inconclusive imaging methods, and also after the second inconclusive imaging technique used to analyse an undefined liver nodule of >1 cm in a patient under surveillance for HCC [3]. Unfortunately, the use of liver biopsy is limited by an elevated rate of false negatives (30%), which increase at a second biopsy (38.9%) [16]. Moreover, liver biopsy has a non-negligible rate of false negative results due to a notable insufficient sampling rate (up to 15%) [26], and it is not free from risks such as bleeding or seeding [27]. In the real clinical scenario, whenever a focal liver lesion shows atypical contrast-enhancement behaviour at imaging, the referring physician typically requires tissue confirmation, not considering that other major and minor limitations include patient constitution, tumour location, and differing operator skills. To date, there are no data regarding the actual feasibility of liver biopsy in patients under surveillance for HCC with an undefined nodule of >1 cm upon imaging. Considering the increasing rate of lesion detection, including lesions in non-cirrhotic livers, and the increase in lesions identified by more refined surveillance schemes, it is necessary to understand the real role of liver biopsy in these scenarios.

The aim of this study was to establish the effective rate of liver biopsy for hepatic lesions with atypical contrast behaviour for HCC in patients under surveillance who are at high risk of developing HCC, and to assess the factors which influence the decision of whether or not to perform tissue sampling.

## 2. Materials and Methods

This was a prospective study approved by the Local Institutional Review Board (protocol number: 216/2018/AOUBo). Written informed consent was obtained from all the patients, and the study was conducted in compliance with the Declaration of Helsinki for clinical studies.

### 2.1. Patient Characteristics and Imaging Technique

This prospective study enrolled patients with atypical nodules under surveillance for HCC, who underwent CECT from September 2018 to September 2019 to further investigate a suspected lesion detected on US.

Contrast-enhanced CT was performed according to the standards of reference recommended by the international guidelines [3,4,28,29,30,31,32].

According to the European Association for the Study of the Liver (EASL), which provides the clinical practice guidelines for the management of HCC [3], a nodule which does not show arterial phase hyperenhancement and wash-out in the portal venous phase of CT is defined as atypical. For a liver nodule in patients under surveillance for HCC to be considered atypical, at least one of the following criteria must be met: (1) arterial phase hyperenhancement without the venous or delayed wash-out; (2) no arterial hyperenhancement, but observed venous or delayed wash-out; (3) hyperdensity in the arterial and delayed phases; (4) hypodensity in the arterial and delayed phases; (5) hyperdensity only in the delayed phase; (6) rim enhancement; (7) hypodensity in the arterial phase and isodensity in the venous and delayed phases; and (8) other.

During the study period, 432 CECTs were performed on patients with liver nodules of >10 mm detected on US, and 25% of the lesions identified were classified as atypical nodules at CECT (Figure 1).

For all atypical nodules, the dimension, liver segment location, and site (superficial in contact with the anterior margin, superficial in contact with the posterior margin, central, or deep behind a fictitious line along the portal axis) were recorded.

Moreover, gender, age, and HCC risk factors were collected for each patient.

### 2.2. Imaging Processing and Analysis

After removing the identification of the patients, all the contrast-enhanced CT studies were saved digitally (on CD-ROM) and were identified with progressive numbers. Subsequently, two CT images were selected by the study coordinator, who chose the most representative images for each lesion: one in the arterial phase, and the other in the delayed phase. Then, the images were inserted into a digital atlas (PowerPoint, Microsoft Corporation, Redmond, WA, USA). These two images were selected with the intent of identifying (or not) the arterialisation and the wash-out of the contrast media, respectively. For each case, a slide with the two images of the lesion (identified by an arrow) in the arterial and the delayed phases was preceded by another slide with an alpha-numeric code. The two slide decks were reordered randomly and were then converted to portable document format (PDF), in a single document, and distributed to the two readers.

After the prospective collection of the data, the CD-ROM and the PDF document were the supports for the present virtual study. The CD-ROM evaluation was used to simulate the diagnostic decision based on the entire imaging study. The PDF document was designed to simulate a possible diagnostic decision, such as that of the feasibility of biopsy, based only on two images, as could often happen during the examination of a report with the principal images attached, or during a multidisciplinary meeting.

One radiologist (Reader 1) with more than ten years of experience in liver imaging and interventional radiology reviewed the PDF document with the two slide decks, with the intent of defining the feasibility of liver biopsy based on the two individual images. Subsequently, after a one-month period to avoid recall bias, Reader 1 reviewed the PDF document for the second time with the same intent, i.e., to assess the intra-observer variability. During both reviews, the reasons for the unfeasibility of biopsy were specified and justified by (a) the location of the lesion (superficial, deep, and vessel proximity) or (b) patient characteristics (i.e., obesity, interposition of the intestinal loop, etc.). Subsequently, Reader 1 evaluated the CD-ROM of each patient with the same intent, namely, to define the feasibility of liver biopsy and to record the reasons if it was not considered possible.

Thereafter, to assess the inter-observer variability, another radiologist (Reader 2) with more than twenty years of experience in liver imaging and interventional radiology blindly and independently reviewed both the PDF document with the two slide decks and the CD-ROM with the entire contrast-enhanced CT study of each patient. Reader 2 was then asked to define the feasibility of liver biopsy during both reviews, as Reader 1 had, and, in the case of unfeasibility, to report the specific reasons, as described with respect to Reader 1.

### 2.3. Statistical Analysis

The continuous variables were expressed as the mean and standard deviation (SD) after assessing for a normal distribution using the Kolmogorov–Smirnov test. The categorical variables were expressed as the number and the percentage. An asymptotic Z-test for determining proportions was used to compare the proportions. A Cox univariate analysis was carried out to assess the degree of association between the unfeasibility of liver biopsy and the dimension (categorised as 10–20 mm, 20–30 mm, and >30 mm), liver segment location, and site (as detailed above). The variables associated with the unfeasibility of liver biopsy through the univariate analysis were tested using the Cox multivariate regression model. The hazard ratios (HRs) and 95% confidence intervals (CIs) were calculated as independent predictors of the unfeasibility of liver biopsy. To assess the agreement between the two radiologists, Cohen’s kappa values (κ) were calculated. The 95% CIs for the Cohen’s kappa index were also reported. Kappa values less than 0 indicated no agreement, 0–0.20 indicated a slight agreement, 0.21–0.40 indicated a fair agreement, 0.41–0.60 indicated a moderate agreement, 0.61–0.80 indicated a substantial agreement, and 0.81–1 indicated an almost perfect agreement. All the tests were two-tailed and a *p*-value of < 0.05 was considered statistically significant. The statistical analyses were carried out using the SPSS 26.0 package (SPSS Inc., Chicago, IL, USA) and STATA v16 (StataCorp LLC, College Station, TX, USA).

## 3. Results

### 3.1. Atypical Liver Nodule Characteristics

A total of 128 atypical liver nodules, consecutively diagnosed in 108 patients, were included in the present study. The mean diameter of the liver nodules was 16.3 ± 5.2 mm, with the majority of nodules belonging to the 10–20 mm-sized group. More than half of them (*n* = 83; 64.8%) were located in the right lobe, mainly in an anterior superficial site. In the majority of cases, the atypical contrast behaviour relied on the absence of contrast enhancement and a hypo- or hypervascular appearance in the portal and the delayed phases (*n* = 79; 61.7%). The characteristics of the liver nodules are outlined in Table 1.

### 3.2. Feasibility of Liver Biopsy

After reviewing the set of single images stored in the PDF document, 76/128 (59.4%) atypical liver nodules were deemed unfeasible for biopsy according to the radiologist with more than 10 years’ experience in liver imaging and in interventional radiology, while the radiologist with more experience considered liver biopsy impossible in 68/128 (53.1%) cases (*p* = 0.604). When all the cases were revised in light of the entire contrast-enhanced CT study, reviewing the CD-ROM one month after reading the PDF document, both percentages slightly decreased (first radiologist = 70/128 (54.7%); second radiologist = 61/128 (47.6%); *p* = 0.591). Between the first and the second readers, no differences were found in assessing the unfeasibility of liver biopsy in terms of the nodule dimension, location, and atypical contrast behaviour, both based on the PowerPoint slide deck (Table 2) and based on the complete CT contrast-enhanced study (data not shown).

### 3.3. Intra- and Inter-Reader Agreements

The intra- and inter-reader agreements regarding the unfeasibility of liver biopsy are reported in Table 3. The intra-reader agreement regarding the single imaging set was substantial (kappa = 0.648 (95% CI = 0.513–0.783)). The inter-reader agreement on the PDF document was slight (kappa = 0.185 (95% CI = 0.021–0.348)); however, it subtly increased to moderate when considering the entire contrast-enhancement CT study (kappa = 0.424 (95% CI = 0.269–0.579)).

### 3.4. Factors Associated with the Unfeasibility of Biopsy 

With respect to the radiologist with more experience in liver imaging, the univariate analysis showed that the nodule size (10–20 mm) and liver segment location (segments six and eight) were negatively and positively associated with the feasibility of liver biopsy, respectively, when the liver nodules were assessed using the single images in the PDF document (Figure 2 and Figure 3). These results were confirmed when these variables were tested in the multivariate analysis. When the atypical liver nodules were assessed using complete contrast-enhanced CT study, only the nodule dimension was selected as a prognostic factor of the unfeasibility of liver biopsy (Table 4 and Table 5). No patient characteristics (i.e., obesity, interposition of the intestinal loop, etc.) were demonstrated to be significative factors associated with the unfeasibility of biopsy.

## 4. Discussion

To the best of the authors’ knowledge, this is the first study which proved the actual feasibility of focal liver lesion biopsy by simulating different clinical scenarios. To date, all the international guidelines recommend establishing pathological proof of all those nodules with atypical contrast behaviour found in a cirrhotic liver, and of all those nodules found in a non-cirrhotic liver without any evidence of other primary distant tumours [3,4,32]. Theoretically, the need for focal liver biopsy in the future will increase due to both the rapidly growing incidence of NAFLD-non-alcoholic steatohepatitis (NAFLD-NASH), which represents a risk factor for HCC in the absence of frank cirrhosis, and the widespread use of MRI performed with hepato-specific contrast media. The multiparametric properties of this imaging technique, together with the use of a cell-specific gadolinium-based contrast agent, has improved MRI sensitivity in detecting liver lesions. Moreover, recent advances in accelerated MRI acquisition (e.g., higher-order parallel imaging, k-space and temporal dimension acceleration, the compressed sensing technique, non-Cartesian acquisition, and simultaneous multi-section imaging) have permitted the implementation of abbreviated MRI protocols for HCC surveillance [33]. The issue of atypical liver nodules, mainly those without arterial vascularisation, the so-called hypovascular nodules, has been investigated in depth in recent years. In their studies, Leoni et al. [34] showed that a non-negligible proportion of liver lesions did not show arterial hypervascularity (39%), highlighting the need for liver biopsy in order to reach a final diagnosis. In fact, 8.3% of these atypical nodules turned out to be HCC, and the percentage was even higher for smaller nodules ranging between 10–20 mm (17%) [28]. The advent of new direct-acting antivirals (DAAs) against HCV, together with the widespread use of MRI performed with hepato-specific contrast agents, has opened a new chapter in the investigation and management of nodules without arterial hypervascularisation [21,22]. In fact, in patients who reached a sustained virological response (SVR) after DAAs, the presence of hypovascular/hypointense liver nodules before treatment represents a risk factor for HCC appearance/relapse, inasmuch as they represent a precursor of malignancy. In addition, both the recurrence risk and survival of patients with successfully treated HCV-related HCC are extremely variable [35,36]. In this context, the need for proven pathology of atypical liver nodules becomes clear in order to offer the patients the best management.

The proportion of atypical liver nodules found in the present study (25%) is in line with those found in previous studies [27,37].

The state-of-the-art guidelines for the management of HCC suggest, if feasible, the biopsy of those nodules which show atypical contrast behaviour after the first or the second imaging. Unfortunately, the rate of the unfeasibility of biopsy is still unknown.

However, according to the present results, liver biopsy was considered to be feasible only in nearly half of the cases, even when the images were read by experienced radiologists. The feasibility was lower when cases were reviewed using a single image presented on a PowerPoint slide deck. This likely simulated the scenario of a multidisciplinary meeting, during which there is often not enough time to correctly evaluate the entire imaging study of each patient. The feasibility rate of liver biopsy increased (52.4%) when the cases were analysed using the entire contrast-enhanced CT study, as the radiologist had the time to analyse and interpret the images. In less expert hands, liver biopsy was deemed to be even less feasible, with poor agreement between the two radiologists when the cases were analysed based on single images (k = 0.185), and only fair agreement when the cases were reviewed based on the entire CT study (k = 0.424), thus demonstrating the poor reproducibility of the final decision as to whether or not to biopsy a lesion. The factors associated with the unfeasibility of biopsy were the lesion dimension and location. Despite the fact that these limitations are easily understood, no clear statements have been made regarding this fact in the published literature, and biopsy has been suggested for all nodules with an atypical vascular pattern on CT or MRI. Smaller nodules, ranging from 10–20 mm, are less prone to being punctured, as they can be faintly seen on US, which, in the majority of cases, is used as a guide for liver biopsy. The segment location may also represent an obstacle [38]. In this study, only segment six was a positive independent prognostic factor for biopsy, probably due to the better visualisation of, and approach to, this portion of the liver through US. Surprisingly, neither the type of atypical vascular behaviour nor the position of the liver nodules (superficial anterior or posterior, and centrohepatic) affected the decision as to whether or not to biopsy a lesion. Although it was not considered in the present analysis, it is also worth remembering that the widespread presence of a metabolic syndrome, namely regarding obese patients with NAFLD-NASH, adds an additional possible limitation to the feasibility of liver biopsy.

The results of the present study have important implications for clinical practice and the management of atypical nodules in cirrhotic patients. In fact, in the current guidelines for the management of HCC [3], the diagnostic flowchart recommends performing a biopsy for atypical nodules after two inconclusive imaging techniques or even after only one inconclusive imaging method. However, as demonstrated in the present study, the feasibility rate of liver biopsy is low and, therefore, the strategy for achieving the correct diagnosis of an atypical nodule in a cirrhotic background differs among different centres. If the present results are confirmed, it will be necessary to offer, in the guidelines, an alternative strategy for biopsy in all those cases where it is foreseen but unfeasible, in order to guarantee an optimal and equal choice for all cirrhotic patients with atypical nodules identified on imaging.

This study had some limitations. First of all, it included patients from a single medical centre. However, this did not represent a limitation according to the aim of the present virtual study. Secondly, a real clinical scenario was simulated, thus probably reducing the reliability of the results. However, the decision concerning the feasibility of biopsy of each atypical nodule was strictly adherent to the principles which are utilised in real clinical practice. Thirdly, the sample size was too small to provide robust evidence and more studies are required to strengthen these findings. Finally, the present data and, in particular, the feasibility of liver biopsy were evaluated only with regard to CT images. Therefore, it is also important to confirm the present results with a dedicated prospective study based on the subsequent US evaluation of the feasibility of liver biopsy.

## 5. Conclusions

This study reported, for the first time, the actual feasibility of liver biopsy of atypical liver nodules in patients with a high risk of developing HCC, which was deemed possible in slightly more than half of the cases, even by experienced radiologists. The unfeasibility of liver biopsy was mainly due to the small size of the lesions and their location. Moreover, there was poor agreement between the radiologists in determining the possibility of whether or not to biopsy a liver lesion. If the results of the present study are confirmed, changing the diagnostic algorithm for those liver lesions which show atypical contrast behaviour upon imaging in patients under surveillance for a high risk of developing HCC should be considered.

## Figures and Tables

**Figure 1 jcm-11-04399-f001:**
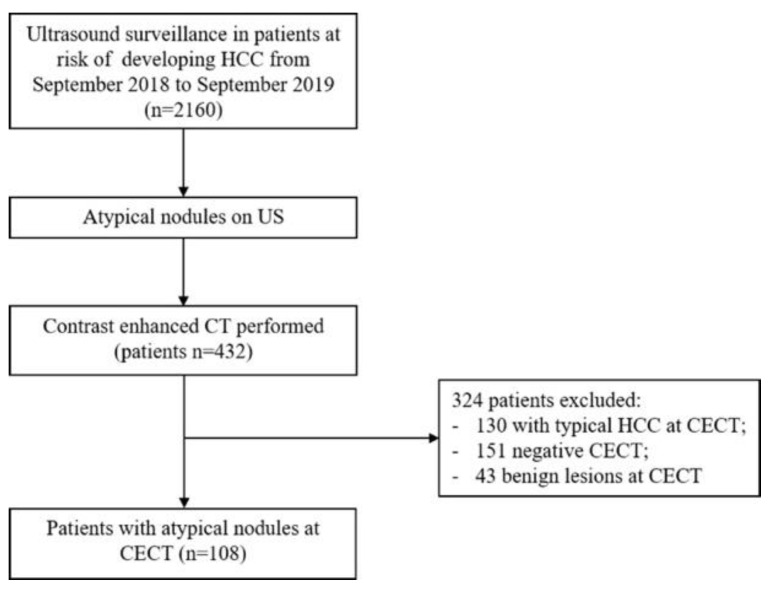
Flow diagram of the patient selection in this study.

**Figure 2 jcm-11-04399-f002:**
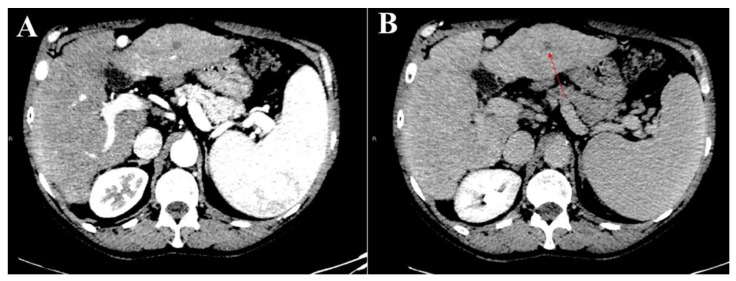
Representative sets of CT images of an atypical liver nodule in segment three in the arterial (**A**) and the delayed (**B**) phases (red arrow) were presented to the readers in PDF format. In particular, the nodule was hypodense in both the arterial and the delayed phases, and its biopsy was considered feasible.

**Figure 3 jcm-11-04399-f003:**
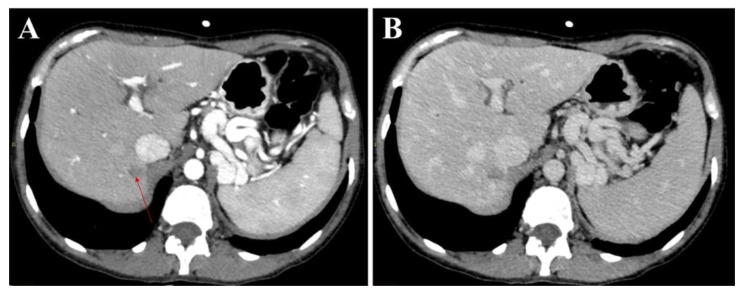
Representative sets of CT images of the same atypical liver nodule in segment seven, which appeared hypodense in both the arterial (**A**) (red arrow) and the delayed (**B**) phases, were presented to the readers in PDF format. In particular, both radiologists considered a liver biopsy of this nodule unfeasible due to its posterior-superior location and its adjacency to the inferior vena cava.

**Table 1 jcm-11-04399-t001:** Characteristics of atypical liver nodules.

Characteristics	Nodules (N° 128)	*p* Value
**Dimension** (mm) [mean ± SD] *Subgroups*	16.3 ± 5.2	
10–20 (*n*; %)	107 (83.6)	ref.
21–30 (*n*; %)	12 (9.3)	<0.001
31–40 (*n*; %)	9 (7.0)	<0.001
**Liver segment**		
1 (*n*; %)	0	<0.001
2 (*n*; %)	16 (12.5)	ref.
3 (*n*; %)	16 (12.5)	1.000
4 (*n*; %)	13 (10.1)	0.554
5 (*n*; %)	19 (14.8)	0.585
6 (*n*; %)	14 (10.9)	0.698
7 (*n*; %)	18 (14.1)	0.713
8 (*n*; %)	32 (25.0)	0.010
**Location**		
Anterior, superficial (*n*; %)	67 (52.3)	ref.
Posterior, superficial (*n*; %)	25 (19.5)	<0.001
Centrohepatic (*n*; %)	28 (21.9)	<0.001
Deep behind a fictitious line of the portal axis (*n*; %)	8 (6.2)	<0.001
**Contrast behaviour**		
Arterial hypervascularisation (*n*; %)	49 (38.3)	ref.
No arterial hypervascularisation (*n*; %)	79 (61.7)	<0.001

SD: standard deviation.

**Table 2 jcm-11-04399-t002:** Comparison of the unfeasibility of biopsy between the first and the second readers based on the PDF document.

Characteristics	First Reader (No Biopsy = 76)	Second Reader (No Biopsy = 68)	*p* Value
**Dimension** (mm) [mean ± SD] *Subgroups*	15.2 ± 5.1	15.3 ± 5.2	0.779
10–20 mm (*n*; %)	66 (86.8)	61 (89.7)	0.578
21–30 mm (*n*; %)	7 (9.2)	4 (5.9)	0.414
31–40 mm (*n*; %)	3 (3.9)	3 (4.4)	1
**Liver segment**			
1 (*n*; %)	0		
2 (*n*; %)	6 (7.9)	9 (13.2)	0.479
3 (*n*; %)	5 (6.6)	7 (10.3)	0.716
4 (*n*; %)	8 (10.5)	5 (7.4)	0.433
5 (*n*; %)	11 (14.5)	12 (17.6)	1
6 (*n*; %)	5 (6.6)	3 (4.4)	0.678
7 (*n*; %)	14 (18.4)	10 (14.7)	0.289
8 (*n*; %)	27 (35.5)	22 (32.4)	0.237
**Location**			
Anterior, superficial (*n*; %)	44 (57.9)	36 (52.9)	0.217
Posterior, superficial (*n*; %)	14 (18.4)	12 (17.6)	0.777
Centrohepatic (*n*; %)	14 (18.4)	16(23.5)	0.789
Deep behind a fictitious line of the portal axis (*n*; %)	4 (5.3)	4 (5.9)	1
**Contrast behaviour**			
Arterial hypervascularisation (*n*; %)	16 (21.0)	18 (26.5)	0.832
No arterial hypervascularisation (*n*; %)	36 (47.4)	42 (61.8)	0.426

SD: standard deviation.

**Table 3 jcm-11-04399-t003:** Inter- and intra-reader agreements.

	Inter-Reader Agreement (95% CI)	Intra-Reader Agreement (95% CI)
**The PDF document**	0.185 (0.021–0.348)	0.648 (0.513–0.783)
**Complete contrast-enhanced CT study**	0.424 (0.269–0.579)	

CI: confidence interval.

**Table 4 jcm-11-04399-t004:** Factors associated with the unfeasibility of liver biopsy based on single images by the second reader.

	Univariate Analysis			Multivariate Analysis		
Variables	HR	95%CI	*p*-Value	HR	95% CI	*p* Value
**Dimension** (mm) *Subgroups*	1.040	0.989–1.094	0.125			
10–20 (*n*; %)	**2.652**	**0.991–7.100**	**0.052**	**3.639**	**1.257–10.381**	**0.016**
21–30 (*n*; %)	0.406	0.116–1.425	0.159			
31–40 (*n*; %)	0.415	0.099–1.739	0.229			
**Liver segment**						
1 (*n*; %)	--					
2 (*n*; %)	1.155	0.402–3.317	0.789			
3 (*n*; %)	0.650	0.226–1.868	0.424			
4 (*n*; %)	0.516	0.159–1.672	0.270			
5 (*n*; %)	1.622	0.594–4.432	0.345			
**6 (*n*; %)**	**0.206**	**0.054–0.777**	**0.020**	**0.220**	**0.057–0.859**	**0.029**
7 (*n*; %)	1.121	0.411–3.053	0.824			
**8 (*n*; %)**	**5.180**	**1.840–14.577**	**0.002**	**2.416**	**0.956–6.049**	**0.060**
**Location**						
Anterior, superficial	1.052	0.525–2.109	0.885			
Posterior, superficial	0.775	0.323–1.859	0.568			
Centrohepatic	1.231	0.529–2.865	0.630			
Deep behind a fictitious line of the portal axis	0.875	0.209–3.662	0.855			

HR: hazard ratio; CI: confidence interval.

**Table 5 jcm-11-04399-t005:** Factors associated with the unfeasibility of liver biopsy based on complete contrast-enhanced CT study by the second reader.

	Univariate Analysis			Multivariate Analysis	
Variables	HR	95% CI	*p* Value	HR	95% CI
**Dimension** (mm) *Subgroups*	**1.057**	**0.999–1.118**	**0.054**		
10–20 (*n*; %)	2.644	0.954–7.332	0.062		
21–30 (*n*; %)	0.518	0.148–1.814	0.303		
31–40 (*n*; %)	0.219	0.058–1.457	0.133		
Liver segment					
1 (*n*; %)	--				
2 (*n*; %)	1.484	0.516–4.262	0.464		
3 (*n*; %)	0.835	0.291–2.399	0.738		
4 (*n*; %)	0.658	0.203–2.134	0.486		
5 (*n*; %)	1.264	0.476–3.353	0.638		
6 (*n*; %)	0.400	0.119–1.350	0.140		
7 (*n*; %)	0.660	0.238–1.828	0.424		
8 (*n*; %)	1.879	0.834–4.235	0.128		
**Location**					
Anterior, superficial	1.009	0.504–2.020	0.980		
Posterior, superficial	0.680	0.280–1.653	0.394		
Centrohepatic	1.128	0.488–2.608	0.779		
Deep behind a fictitious line of the portal axis	1.905	0.436–8.331	0.392		

HR: hazard ratio; CI: confidence interval.

## Data Availability

All data generated or analyzed during this study are included in this article. Further enquiries can be directed to the corresponding author.

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
