# Peer review of "The Feasibility of Liver Biopsy for Undefined Nodules in Patients under Surveillance for Hepatocellular Carcinoma: Is Biopsy Really a Useful Tool?"

_jcm, 2022, doi:10.3390/jcm11154399_

Round 1

Reviewer 1 Report

This is a nice study addressing determining biopsy feasibility for atypical liver nodules in patients under surveillance for hepatocellular carcinoma (HCC), and factors influencing the decision to make liver biopsy.
Overall, 128 atypical liver nodules in 108 patients under surveillance for HCC who underwent CT were 19 included. Intra- and inter-observer variability of two experienced radiologists was assessed. 
They found that when reviewing the digital atlas, 76 (59.4%) 24 and 68 (53.1%) nodules were deemed unfeasible for biopsy by the less-experienced radiologist 25 (Reader 1) and the more-experienced radiologist (Reader 2), respectively. When reviewing the entire CT study, both percentages decreased slightly [Reader 1 (70/128 (54.7%); Reader 2=61/128 27 (47.6%); P=0.591]. The intra-reader agreement on PDF was substantial [k=0.648 (95%CI=0.513- 28 0.783)]; the inter-reader agreement on PDF was slight [k=0.185 (95%CI=0.021-0.348)] and moderate 29 on the entire CT study [k=0.424 (95%CI=0.269-0.579)].
Of interest, when assessing the digital atlas, nodule size (10-20 30 mm) and location in segments 6 and 8 were negatively and positively associated with liver biopsy feasibility, respectively. When assessing the CD-ROM, only the nodule dimension was associated with liver biopsy unfeasibility.
They concluded that the unfeasibility of liver biopsy is mainly due to the small size of the lesions and their location.
In my opinion, the study has a clinical impact. I have only minor points to suggest:
-when the authors discuss the problem of HCC related to direct-acting antivirals (DAAs), they should also recall that recurrence risk and survival are extremely variable in patients with successfully treated HCV-related HCC, as previously demonstrated in previous studies (A meta-analysis of single HCV-untreated arm of studies evaluating outcomes after curative treatments of HCV-related hepatocellular carcinoma. Liver International 2017;37(8):1157-1166Hepatocellular carcinoma recurrence in patients with curative resection or ablation: impact of HCV eradication does not depend on the use of interferonAlimentary Pharmacology and Therapeutics 2017;45:160-168).

Author Response

Dear Reviewer, thank you for your kind appreciations. As you suggested, we added a brief consideration regarding the extremely variable recurrence risk and survival in patients with successfully treated HCV-related HCC, also citing the suggested references.

Reviewer 2 Report

This is a prospective study that looked into the concordance rates of feasibility of resection between two radiologists and also one radiologist two different times. 

Overall, this is an interesting study but clarity on the intentions of the study and link between the study results and conclusion is needed. Authors designed the study well but it is difficult to come to conclusion 

Author Response

Dear Reviewer, thank you for your comment. The present study aims to provide the feasibility rate of biopsy for atypical liver nodules in patients under surveillance for hepatocellular carcinoma, a data which is currently not known in the literature. The result that emerged from our study has important clinical implications as biopsy feasibility does not reach a sufficient rate. In this scenario, we believe that the EASL guidelines should be implemented with these new considerations, introducing a second alternative strategy in all circumstances in which biopsy is proposed.

Round 2

Reviewer 2 Report

Please answer at least some of the questions we posed to re-consider the submission

Author Response

Dear Reviewer, thank you for your comment. The present study aims to provide the feasibility rate of biopsy for atypical liver nodules in patients under surveillance for hepatocellular carcinoma, a data which is currently not known in the literature. The result that emerged from our study has important clinical implications as biopsy feasibility does not reach a sufficient rate. In this scenario, we believe that the EASL guidelines should be implemented with these new considerations, introducing a second alternative strategy in all circumstances in which biopsy is proposed. We have now reported these considerations also in the paper, in order to facilitate and clarify the link between the results presented and the conclusions.

We hope that the manuscript in the present form, improved according with your suggestions, may reach sufficient priority to be published in Journal of Clinical Medicine.

Kind regards